# The Ypk1 protein kinase signaling pathway is rewired and not essential for viability in *Candida albicans*

**Bernardo Ramírez-Zavala**[ID]**, Ines Krüger, Andreas Wollner, Sonja Schwanfelder, Joachim Morschhäuser**[ID]*****

Institute of Molecular Infection Biology, University of Würzburg, Würzburg, Germany

* joachim.morschhaeuser@uni-wuerzburg.de

**Data Availability Statement:** The authors confirm that all data underlying the findings are fully available without restriction. All relevant data are

## Abstract

Protein kinases are central components of almost all signaling pathways that control cellular activities. In the model organism *Saccharomyces cerevisiae*, the paralogous protein kinases Ypk1 and Ypk2, which control membrane lipid homeostasis, are essential for viability, and previous studies strongly indicated that this is also the case for their single ortholog Ypk1 in the pathogenic yeast *Candida albicans*. Here, using FLP-mediated inducible gene deletion, we reveal that *C. albicans ypk1Δ* mutants are viable but slow-growing, explaining prior failures to obtain null mutants. Phenotypic analyses of the mutants showed that the functions of Ypk1 in regulating sphingolipid biosynthesis and cell membrane lipid asymmetry are conserved, but the consequences of *YPK1* deletion are milder than in *S. cerevisiae*. Mutational studies demonstrated that the highly conserved PDK1 phosphorylation site T548 in its activation loop is essential for Ypk1 function, whereas the TORC2 phosphorylation sites S687 and T705 at the C-terminus are important for Ypk1-dependent resistance to membrane stress. Unexpectedly, Pkh1, the single *C. albicans* orthologue of Pkh1/Pkh2, which mediate Ypk1 phosphorylation at the PDK1 site in *S. cerevisiae*, was not required for normal growth of *C. albicans* under nonstressed conditions, and Ypk1 phosphorylation at T548 was only slightly reduced in *pkh1Δ* mutants. We found that another protein kinase, Pkh3, whose ortholog in *S. cerevisiae* cannot substitute Pkh1/2, acts redundantly with Pkh1 to activate Ypk1 in *C. albicans*. No phenotypic effects were observed in cells lacking Pkh3 alone, but *pkh1Δ pkh3Δ* double mutants had a severe growth defect and Ypk1 phosphorylation at T548 was completely abolished. These results establish that Ypk1 is not essential for viability in *C. albicans* and that, despite its generally conserved function, the Ypk1 signaling pathway is rewired in this pathogenic yeast and includes a novel upstream kinase to activate Ypk1 by phosphorylation at the PDK1 site.

## Author summary

Protein kinases are key components of cellular signaling pathways, and elucidating the specific roles of individual kinases is important to understand how organisms adapt to

within the paper and its Supporting Information files.

**Funding:** This study was funded by the German Research Foundation (DFG) through the TRR 124 FungiNet "Pathogenic fungi and their human host: Networks of Interaction", DFG project number 210879364, Project C2 to JM. Publication of the work was supported by the Open Access Publication Fund of the University of Würzburg. The funders had no role in study design, data collection and analysis, decision to publish, or preparation of the manuscript.

**Competing interests:** The authors have declared that no competing interests exist.

changes in their environment. The protein kinase Ypk1 is highly conserved in eukaryotic organisms and crucial for the maintenance of cell membrane homeostasis. It was previously thought that Ypk1 is essential for viability in the pathogenic yeast *Candida albicans*, as in the model organism *Saccharomyces cerevisiae*. Here, by using forced, inducible gene deletion, we reveal that *C. albicans* mutants lacking Ypk1 are viable but have a strong growth defect. The phenotypes of the mutants indicate that the known functions of Ypk1 are conserved in *C. albicans*, but loss of this kinase has less severe consequences than in *S. cerevisiae*. We also unravel the puzzling previous observation that *C. albicans* mutants lacking the Ypk1-activating kinase Pkh1, which is essential in *S. cerevisiae*, have no obvious growth defects. We show that the protein kinase Pkh3, which has not previously been implicated in the Ypk1 signaling pathway, can substitute Pkh1 and activate Ypk1 in *C. albicans*. These findings provide novel insights into this conserved signaling pathway and how it is rewired in a human-pathogenic fungus.

## Introduction

Protein kinases are key components of many signaling pathways that regulate cellular activities and the responses of cells to external signals. Deciphering the functions of individual protein kinases is therefore important to understand the regulatory networks that control the behavior of organisms and how they adapt to changes in their environment. The yeast *Candida albicans* is a harmless colonizer of different mucosal surfaces in healthy humans, but it can also become a pathogen and infect almost all body locations when host defenses are compromised. To enable systematic investigations of the role of protein kinases in the biology and pathogenicity of *C. albicans*, we have recently generated a comprehensive library of protein kinase deletion mutants of the wild-type reference strain SC5314 [1,2]. The *C. albicans* genome contains 108 genes encoding known or predicted protein kinase catalytic subunits, and we had successfully constructed homozygous deletion mutants for 86 of these genes. For the remaining 22 protein kinase genes, only heterozygous mutants were obtained, indicating that these are essential for viability or the selection conditions prevented the recovery of null mutants. Among the latter group was *YPK1* (orf19.399). Although either of the two *YPK1* alleles could be deleted in a first round of transformation, none of 92 analyzed second-round transformants of the two types of heterozygous mutants had deleted the remaining wild-type allele. This was not surprising, since homozygous *ypk1* mutants were also not recovered in a previous effort by other researchers to generate a protein kinase insertion mutant library [3]. Furthermore, a genome-wide *in vivo* transposon mutagenesis study with a haploid *C. albicans* strain, combined with machine learning, strongly suggested that *YPK1* is an essential gene [4].

The model yeast *Saccharomyces cerevisiae* contains two partially redundant paralogous protein kinases, Ypk1 and Ypk2, which regulate plasma membrane lipid and protein homeostasis [5]. Mutants lacking Ypk1 display a slow-growth phenotype, especially at low temperatures, whereas *ypk2Δ* mutants show wild-type growth, but *ypk1Δ ypk2Δ* double mutants are inviable [6–8]. Ypk1/2 activity requires phosphorylation in their activation loop by the paralogous, plasma membrane-localized protein kinases Pkh1 and Pkh2 and at multiple sites in their C-termini by TOR complex 2 (TORC2) [6,8–13]. Two important downstream targets of Ypk1/2 are the paralogous protein kinases Fpk1 and Fpk2, whose function is inhibited by Ypk1/2-mediated phosphorylation [14]. Fpk1 and Fpk2 phosphorylate and thereby stimulate flippases that translocate aminoglycerophospholipids from the outer to the inner leaflet of the plasma membrane [14–16]. By inhibiting Fpk1 and Fpk2, Ypk1/2 therefore downregulate the

rate of inward aminophospholipid translocation to adjust the lipid distribution in the plasma membrane [14]. Furthermore, TORC2-activated Ypk1/2 increase sphingolipid biosynthesis by phosphorylating and thereby inhibiting the negative regulators of serine-palmitoyl transferase, Orm1 and Orm2, and by stimulating ceramide synthase, to maintain membrane homeostasis in response to stress [9,13,17–19]. A comprehensive description of Ypk1/2-dependent signaling pathways can be found in two recent review articles [5,20].

To gain insight into why Ypk1 might be essential for viability in *C. albicans*, we decided to generate conditional *ypk1Δ* mutants by forced, inducible gene deletion. This strategy was previously established in our lab to generate conditional-lethal *cdc42Δ* and *bem1Δ* mutants of the auxotrophic *C. albicans* laboratory strain CAI4 [21] and recently modified to construct *snf1Δ* mutants of the wild-type reference strain SC5314 [22]. In this approach, a functional copy of the putative essential gene, which is flanked by target sequences of the site-specific recombinase FLP, is inserted at an ectopic site in the genome of heterozygous mutants. The second endogenous allele of the target gene can then be deleted. Finally, the *ecaFLP* (enhanced *Candida*-adapted *FLP*) gene, encoding a mutated version of the FLP recombinase with enhanced activity, is integrated under the control of the tightly regulated and efficiently inducible *SAP2* promoter to generate the desired conditional mutants. Passage in *SAP2*-inducing medium results in the excision of the FLP-deletable gene copy in the vast majority of the cells to produce an almost pure population of null mutants. Plating of these cells and incubation under any desired conditions will provide definite proof of whether the gene is essential (no colony formation) or not. Furthermore, the phenotype and physiological activities of the null mutants can be studied before cell death to obtain clues about the essential biological functions of the encoded protein.

## Results

### *C. albicans ypk1Δ* mutants are viable and have a slow-growth phenotype

To generate conditional *ypk1Δ* mutants in which *YPK1* can be deleted by inducible, FLP-mediated excision from the genome, we followed the modified procedure that we recently described for the construction of conditional *snf1Δ* mutants [22]. A cassette containing the *YPK1* gene and its upstream and downstream sequences, which was flanked by direct repeats of the FLP recombination target sequence *FRT* and also contained the hygromycin resistance marker *HygB* for the selection of transformants, was integrated at the *ADH1* locus of two independently generated heterozygous *YPK1/ypk1Δ* mutants [1]. The second endogenous *YPK1* allele was then replaced by the *SAT1* flipper cassette in these strains. After induction of *FLP* expression to remove the *SAT1* flipper cassette and plating of the cells on YPD medium, we observed two types of colonies, large colonies showing wild-type growth and small, slowly growing colonies. Southern hybridization analysis of several clones of each class showed that the former (SCYPK1M6) had excised the *SAT1*-flipper cassette and retained the FLP-deletable *YPK1* copy, as expected, while the latter (SCYPK1M5) had simultaneously also excised the ectopically integrated *YPK1* copy, suggesting that loss of Ypk1 is not lethal but causes a slow-growth phenotype (Fig 1A). To verify that the slowly growing clones were not suppressor mutants, we proceeded with the construction of the intended conditional mutants by introducing the *ecaFLP* gene under the control of the inducible *SAP2* promoter into the strains in which the ectopically integrated *YPK1* was the only remaining copy of the gene. After passaging the resulting conditional mutants (SCYPK1M7) in *SAP2*-inducing medium and plating on YPD plates, we obtained a homogeneous population of small colonies that had lost the FLP-deletable *YPK1* copy (SCYPK1M8), demonstrating that *ypk1Δ* mutants were indeed viable but growing slowly (Fig 1B). Knowing this phenotype, we undertook a new effort to generate

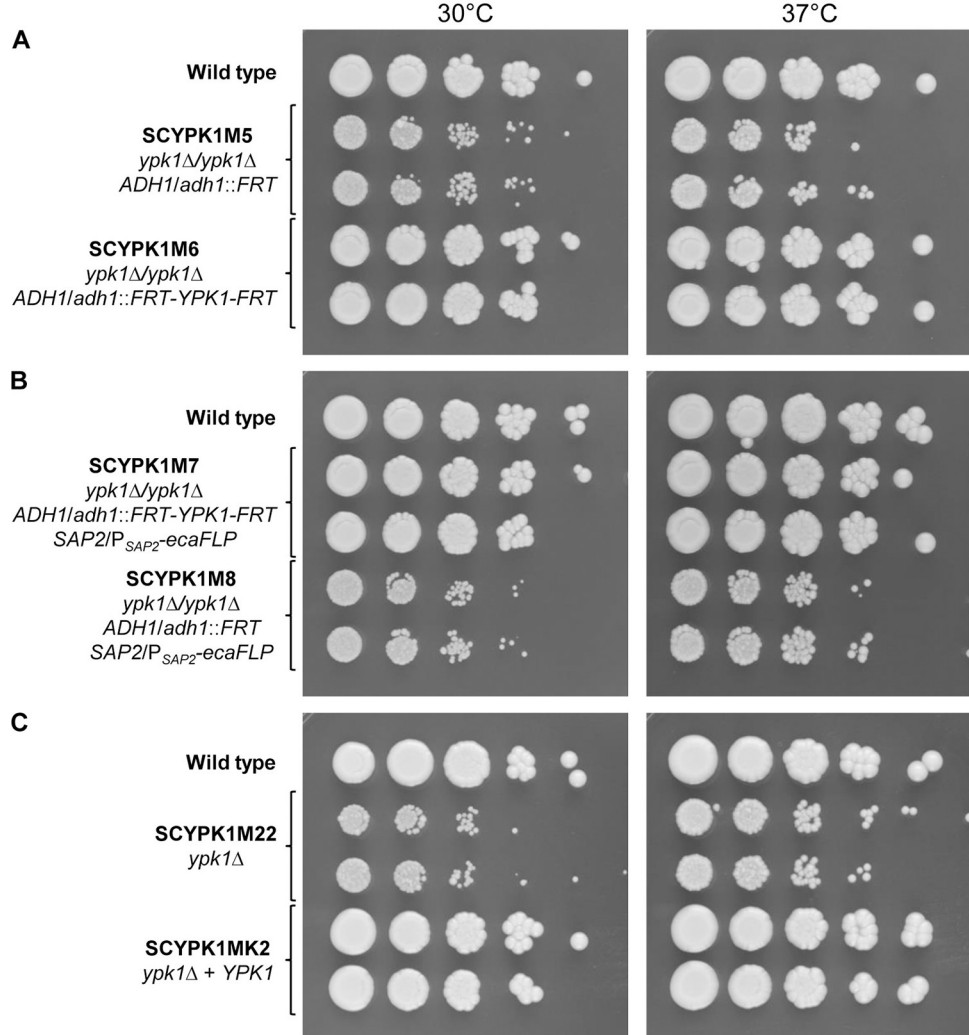

**Fig 1. *C. albicans ypk1Δ* mutants are viable.** Overnight cultures of the wild-type strain SC5314, various *ypk1Δ* mutants (SCYPK1M5, SCYPK1M8, and SCYPK1M22), and corresponding control strains (SCYPK1M6, SCYPK1M7, and SCYPK1MK2) were serially diluted, spotted on YPD agar plates, and incubated for 48 h at 30˚C or 37˚C. Both independently generated series of mutants are shown.

homozygous *ypk1Δ* mutants in the "traditional" way, by deleting the second endogenous *YPK1* allele in the heterozygous mutants using the *SAT1* flipper cassette. Analysis of transformants exhibiting a slow-growth phenotype confirmed deletion of the second *YPK1* allele (SCYPK1M22), and reintroduction of a functional *YPK1* copy at its original locus (SCYPK1MK2) restored wild-type growth (Fig 1C). Subsequent experiments were performed with the latter *ypk1Δ* mutants and complemented derivatives.

## Phenotypes of *ypk1Δ* mutants

The function of Ypk1 and its orthologs is highly conserved in eukaryotes [5]. We therefore tested whether the *C. albicans ypk1Δ* mutants displayed similar phenotypes to those of *S. cerevisiae* mutants with reduced Ypk1/2 function. In *S. cerevisiae*, Ypk1/2 are required to counteract sphingolipid depletion, and *ypk1Δ* mutants are hypersensitive to the sphingolipid biosynthesis inhibitor myriocin [23,24]. We found that at a low myriocin concentration that

did not affect growth of the wild type, growth of the *C. albicans ypk1Δ* mutants was virtually abolished, indicating that they were hypersensitive to the inhibitor (Fig 2A). This was also corroborated in a disk diffusion assay, which demonstrated the hypersensitivity of the *ypk1Δ* mutants by the strongly increased halo around the disks with myriocin, irrespective of their generally slow growth (S1 Fig). In *S. cerevisiae*, Ypk1 is required for efficient uptake of fatty acids, and in contrast to wild-type cells, *ypk1Δ* mutants cannot utilize exogenously supplied fatty acids for growth in the presence of the fatty acid biosynthesis inhibitor cerulenin [25]. To test whether this function of Ypk1 is conserved in *C. albicans*, we compared growth of the wild type and *ypk1Δ* mutants on plates with and without fatty acids and cerulenin. As can be seen in Fig 2B, growth of the *ypk1Δ* mutants was not further reduced by cerulenin when fatty acids were present in the medium, indicating that fatty acid uptake does not depend on Ypk1 in *C. albicans*. *S. cerevisiae ypk1Δ* mutants have a defect in endocytosis [26], and this is also the basis of their impaired uptake of fatty acids from the environment [25]. The *C. albicans ypk1Δ* mutants were able to internalize the dye FM4-64 and transport it to the vacuole, indicating that they do not have a general endocytosis defect (Fig 2C). In *S. cerevisiae*, Ypk1/2 are required for actin polarization in small/medium sized buds [8,27]. Phalloidin staining showed that actin polarization to the buds, as seen in the wild type and complemented strains, was lost in the *C. albicans ypk1Δ* mutants, and no actin cables were seen in the mother cells. (Fig 2D). Collectively, these results showed that despite a generally conserved function of Ypk1, absence of this kinase has less severe effects in *C. albicans* than loss of its orthologs in *S. cerevisiae*, which may explain the viability of the *ypk1Δ* mutants.

## Orf19.223 encodes the protein kinase Fpk1, a downstream target of Ypk1

As explained in the introduction, the paralogous protein kinases Fpk1 and Fpk2 are downstream targets of Ypk1 in *S. cerevisiae*, and Ypk1-mediated phosphorylation inhibits their function [14]. A putative ortholog of these proteins in *C. albicans* is encoded by the uncharacterized orf19.223. Similar to Fpk1 of *S. cerevisiae*, the predicted orf19.223 gene product contains three consensus Ypk1 phosphorylation motifs (R-x-R-x-x-S/T) [6,14] in its N-terminal half, RNRTRSIS (amino acids 60–67 with two overlapping motifs), RTRSAS (amino acids 134–139), and RPRTYT (amino acids 358–363). Therefore, and because of the results reported below, we refer to orf19.223 as *FPK1* in our present study.

In contrast to *ypk1Δ* mutants, *S. cerevisiae* mutants with a catalytically inactive Fpk1 [14] and *fpk1Δ fpk2Δ* double mutants [28] are hyperresistant to myriocin, because Fpk1/2 activate flippases [14] and myriocin is taken up into the cells by a flippase-dependent mechanism [28]. In line with this, we found that *C. albicans fpk1Δ* mutants were also hyperresistant to myriocin, and this phenotype was reverted after integrating an HA-tagged *FPK1* copy at the original locus in the mutants (Fig 3A). If an inability of *ypk1Δ* mutants to inhibit Fpk1 activity contributes to their growth defect, as is the case in *S. cerevisiae* [14], deletion of *FPK1* in cells lacking Ypk1 might ameliorate growth. Indeed, we found that the growth of *ypk1Δ fpk1Δ* double mutants at 37°C was improved compared to that of *ypk1Δ* single mutants (Figs 3B and S2). An increased uptake of the inhibitor due to unrestrained Fpk1-mediated activation of flippases may also contribute to the myriocin hypersensitivity of the *ypk1Δ* mutants (see Fig 2A). As explained in the introduction, in *S. cerevisiae* Ypk1/2 are required for the activities of serine-palmitoyl transferase and ceramide synthase. The still reduced growth of the *ypk1Δ fpk1Δ* double mutants is therefore likely due to a failure to promote sphingolipid synthesis.

To obtain additional evidence for the functional relationship between Ypk1 and Fpk1, we compared Fpk1 levels in wild-type and *ypk1Δ* strains in which one of the endogenous *FPK1* alleles was replaced by an HA-tagged copy. While we did not observe differences in the band

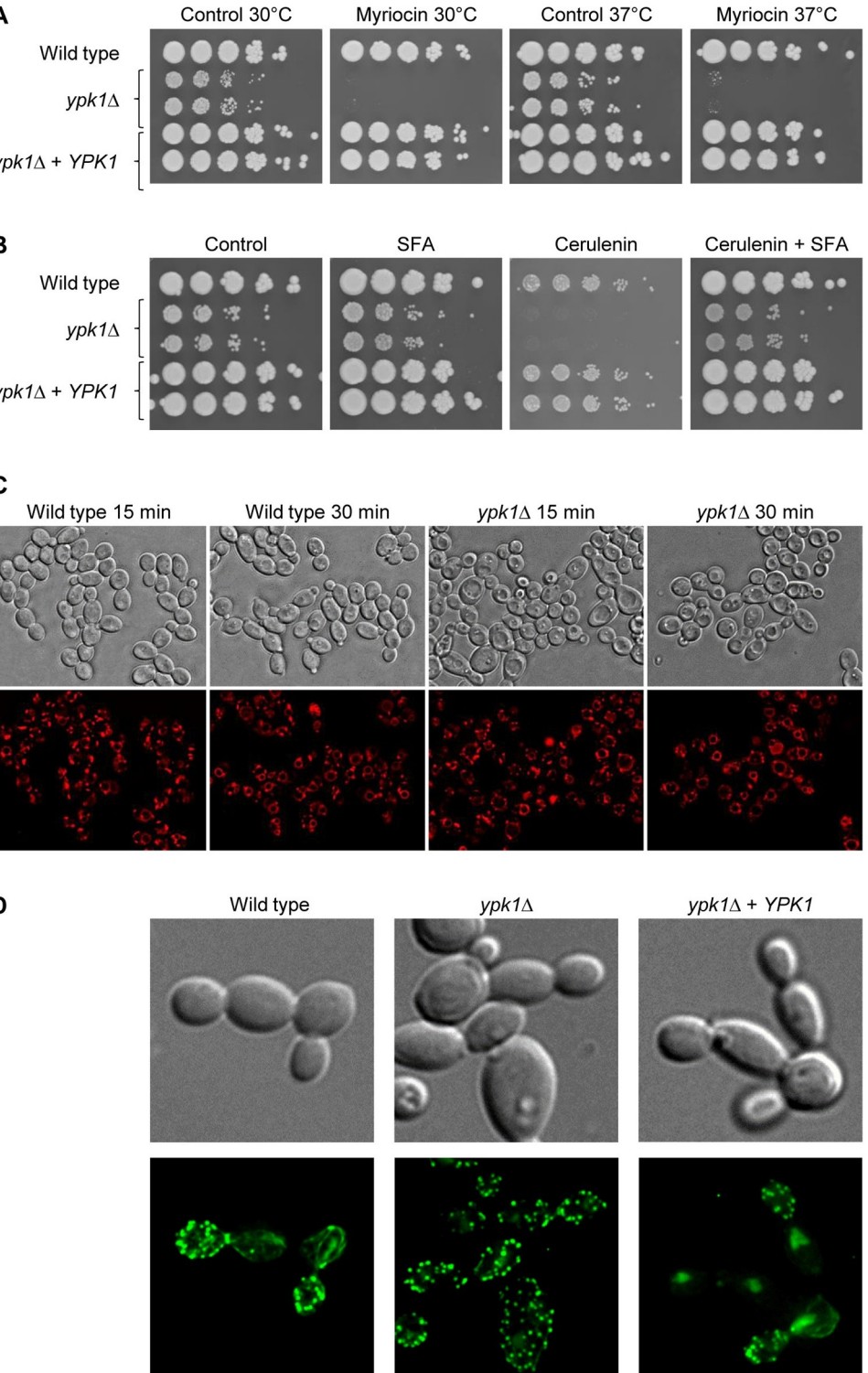

**Fig 2. Phenotypes of *C. albicans ypk1Δ* mutants.** (A) Overnight cultures of the wild-type strain SC5314, *ypk1Δ* mutants, and complemented strains were serially diluted and spotted on YPD agar plates without or with 31.2 ng/ml myriocin. Plates were incubated for 48 h at 30˚C or 37˚C. (B) The same strains were spotted on plates containing saturated fatty acids (SFA, 0.5 mM each of stearic acid and palmitic acid), 10 mM cerulenin, or both cerulenin and SFA and incubated for 48 h at 37˚C. Control plates in (A) and (B) contained the solvent DMSO. Both independently generated series of mutants are shown. (C) FM4-64 uptake in wild-type and *ypk1Δ* cells grown at 30˚C. (D) Actin

staining of the indicated cells with phalloidin. Cells in (C) and (D) were imaged by differential interference contrast (top panels) and fluorescence microscopy (bottom panels) using appropriate filters.

patterns on a Western blot, the levels of HA-tagged Fpk1 were increased in *ypk1Δ* mutants compared to those in wild-type cells, in line with a negative effect of Ypk1 on Fpk1 (Fig 3C, left panels; the signal intensity of the main Fpk1 bands was 1.7-fold higher than in the wild type). Since Fpk1/2 phosphorylate Ypk1 in *S. cerevisiae*, resulting in electrophorectic mobility shifts [14], we also compared Ypk1 levels and phosphorylation patterns in wild-type and *fpk1Δ* strains in which one of the endogenous *YPK1* alleles was replaced by a 3xMyc-tagged copy. Similar amounts of Myc-tagged Ypk1 were detected in wild-type and *fpk1Δ* mutants, but the

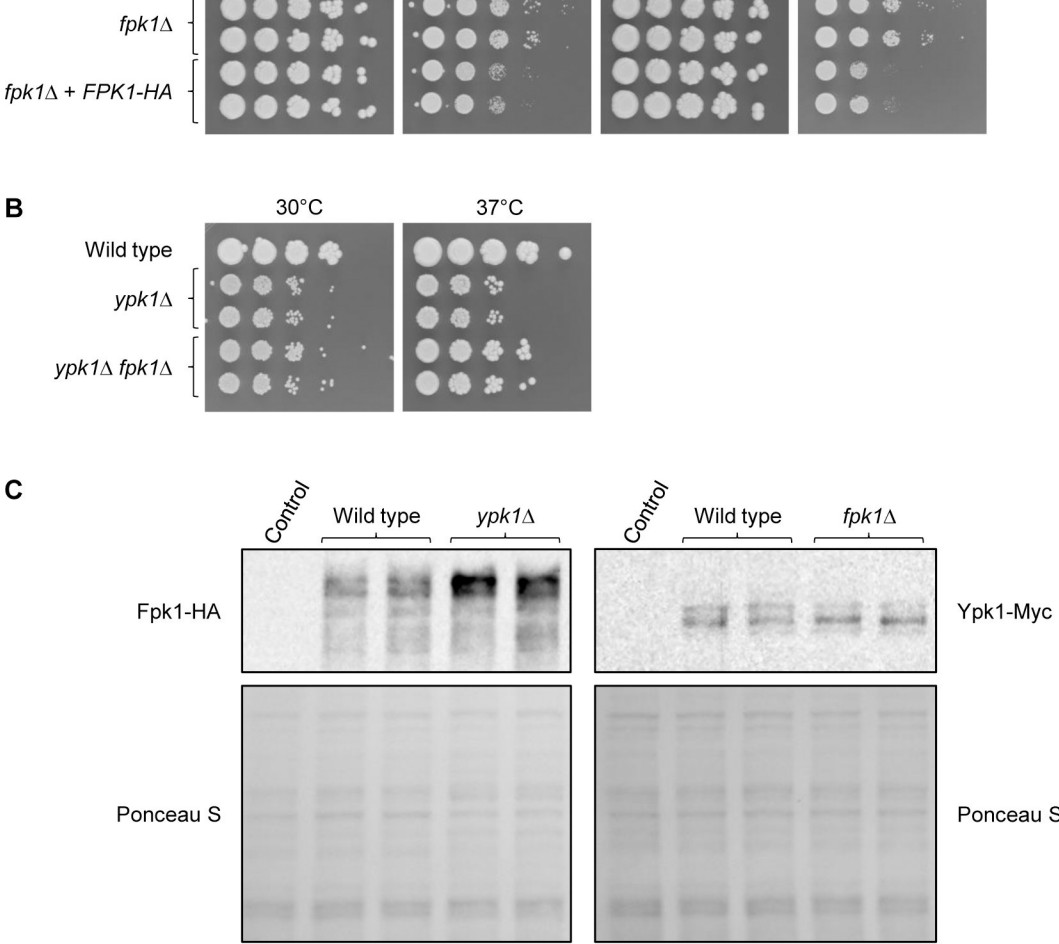

**Fig 3. Fpk1 is a downstream target of Ypk1.** (A) Overnight cultures of the wild-type strain SC5314, *fpk1Δ* mutants, and complemented strains were serially diluted and spotted on YPD agar plates containing 2 μg/ml myriocin or the solvent DMSO (control). Plates were incubated for 48 h at 30°C or 37°C. (B) Serial dilutions of the wild-type strain SC5314, *ypk1Δ* mutants, and *ypk1Δ fpk1Δ* double mutants were spotted on YPD plates and grown for 48 h at 30°C or 37°C. (C) Detection of HA-tagged Fpk1 and Myc-tagged Ypk1 in the indicated strains. Protein extracts were prepared from log-phase cells grown in YPD at 30°C, separated on Phos-tag gels, and analysed on Western blots with anti-HA (left) and anti-Myc (right) antibodies. The untagged parental strain SC5314 served as negative control and Ponceau S staining was used to confirm equal loading. Both independently generated series of mutants are shown in (A) to (C).

relative levels of a lower-mobility form of Ypk1 were decreased in the mutants, indicating that Fpk1 affects the phosphorylation state of Ypk1 (Fig 3C, right panels; the ratio of the signal intensities of the upper band to the middle band was 0.5-fold lower than in the wild type). Altogether, these results provide evidence that Fpk1, encoded by orf19.233, is a downstream target that is inhibited by Ypk1, similar to the situation in *S. cerevisiae*.

## Phosphorylation at the PDK1 site is essential for Ypk1 function in *C. albicans*

The paralogous upstream activating kinases Pkh1 and Pkh2 are essential for Ypk1/2 function and, therefore, viability in *S. cerevisiae* [6,29]. *C. albicans* contains a single *PKH1/2* homolog which we have referred to as *PKH1* in our protein kinase deletion mutant library [1], in accord with an earlier report [3] (the current standard name in the *Candida* Genome Database [30] is *PKH2*). Interestingly, our *pkh1Δ* mutants showed normal growth (see also below), in line with previously reported phenotypes of *pkh1* mutants generated from auxotrophic *C. albicans* laboratory strains [3,31,32]. Since Pkh1/2-mediated phosphorylation at the conserved PDK1 site (named after the mammalian homolog of Pkh1/2) in the activation loop is essential for Ypk1 function in *S. cerevisiae* [24], the normal growth of *pkh1Δ* mutants, as opposed to the slow-growth phenotype of *ypk1Δ* mutants, suggested that phosphorylation at its PDK1 site might be less important for Ypk1 function in *C. albicans* and that phosphorylation by TORC2 has a more critical role. To address this question, we replaced the remaining *YPK1* allele in two independent heterozygous *YPK1/ypk1Δ* mutants (allele 2 in strain A, allele 1 in strain B) by HA-tagged wild-type and mutated *YPK1* copies in which the codons for threonine 548 (the PDK1 site) or the TORC2 target sites serine 687 in the turn motif (TM) or threonine 705 in the hydrophobic motif (HM) were replaced by an alanine codon, which would prevent phosphorylation of Ypk1 at these sites.

On rich YPD medium, the strains containing the HA-tagged *YPK1*^S687A allele grew almost as well as the strains containing the HA-tagged wild-type *YPK1* or the parental wild-type strain SC5314 and heterozygous *YPK1/ypk1Δ* mutants. In contrast, the strains with the *YPK1*^T548A allele exhibited the same severe growth defect as the *ypk1Δ* null mutants, while the strains with the *YPK1*^T705A allele showed an intermediate phenotype (Fig 4A). Western blotting demonstrated that Ypk1^T548A and Ypk1^T705A were produced in even higher amounts than wild-type Ypk1, likely due to an increased demand for Ypk1 activity in cells with compromised Ypk1 function (Fig 4B). In the presence of a low concentration of myriocin, only the strains with a wild-type *YPK1* could still grow, whereas growth of all strains containing mutated *YPK1* alleles was completely inhibited (Fig 4A). The latter result indicates that Ypk1 phosphorylation by TORC2 at the TM (S687) and HM (T705) sites is essential to counteract stress caused by sphingolipid biosynthesis inhibition. The fact that an intact PDK1 site (T548) is indispensable for Ypk1 function, although cells lacking the presumed activating kinase Pkh1 grow normally under unstressed conditions, suggested that an additional/alternative protein kinase mediates Ypk1 phosphorylation at T548 in *C. albicans*.

## Pkh1 and Pkh3 have a redundant function in the Ypk1 signaling pathway in *C. albicans*

In the *Candida* Genome Database, *PKH1* (orf19.5224) is described as encoding a 944 amino acid protein, although the corresponding ORF is not followed by a stop codon. In the current assembly 22, a stop codon comes 9 nucleotides later, predicting a 947 amino acid protein. When constructing a complementation plasmid for our *pkh1Δ* mutants, sequencing showed that both cloned *PKH1* alleles did not contain this stop codon and orf19.5224 was fused in

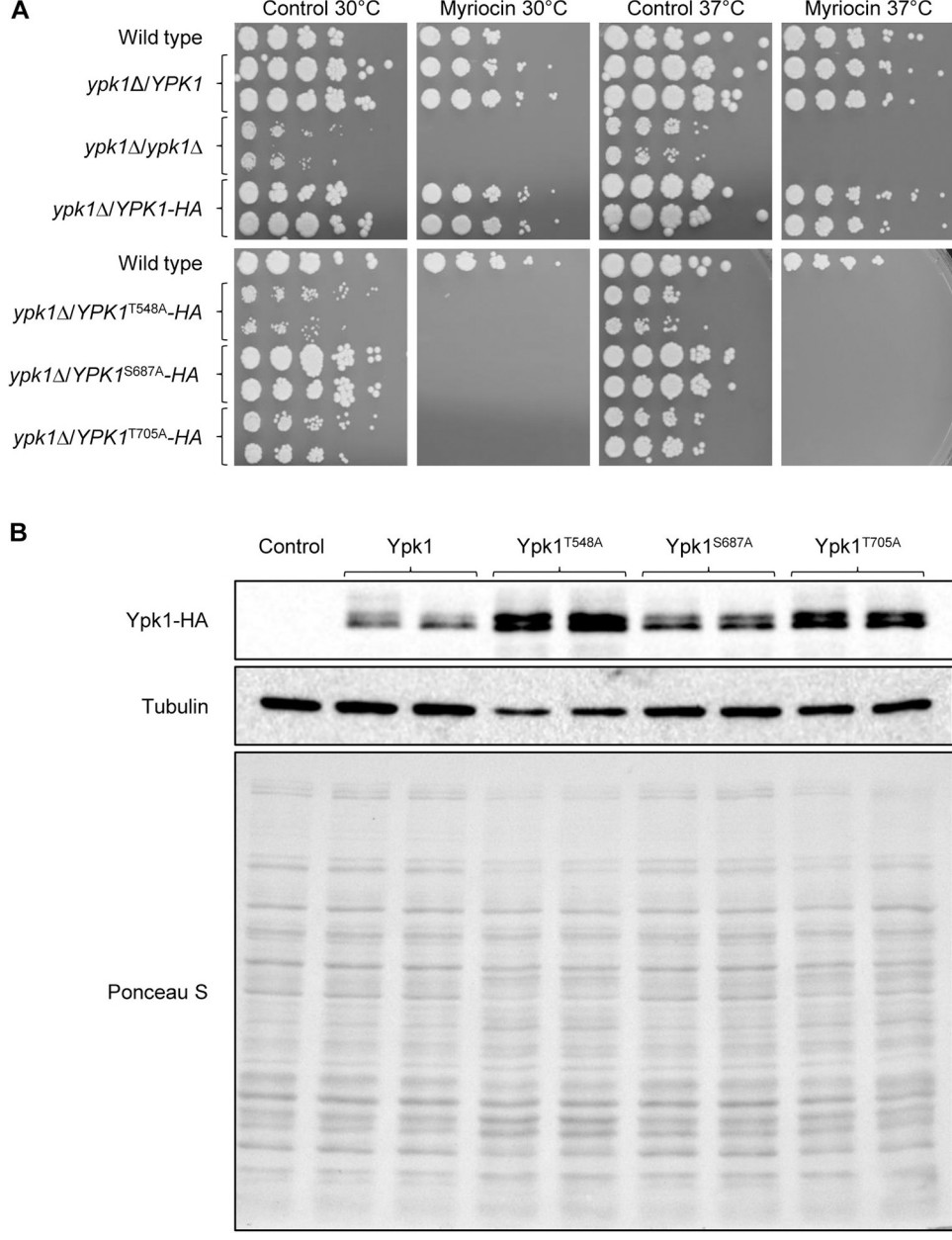

**Fig 4. Growth and myriocin sensitivity of *C. albicans* strains expressing mutated *YPK1* alleles.** (A) Overnight cultures of strains with the indicated genotypes were serially diluted and spotted on YPD agar plates containing 0.125 μg/ml myriocin or the solvent DMSO (control). Plates were incubated for 48 h at 30˚C or 37˚C. Strains in the top and bottom panels were grown on the same plate and the photographs arranged accordingly for clarity of presentation. (B) Detection of HA-tagged wild-type and mutated Ypk1 proteins by Western blotting with an anti-HA antibody. Protein extracts were prepared from log-phase cells grown in YPD at 30˚C. The untagged parental strain SC5314 served as negative control. The blot was also reprobed with an anti-tubulin antibody and stained with Ponceau S as a loading control. Results for both independently generated series of mutants are shown in (A) and (B).

frame via the intervening sequence with the downstream orf19.5225 to encode a predicted 1153 amino acid protein (Fig 5A, see S3 Fig for sequence comparisons). To verify that the *PKH1* coding sequence indeed reaches from the start codon of orf19.5224 to the stop codon of orf19.5225, we fused a 3xHA tag to the last codon of orf19.5225 and inserted the HA-tagged

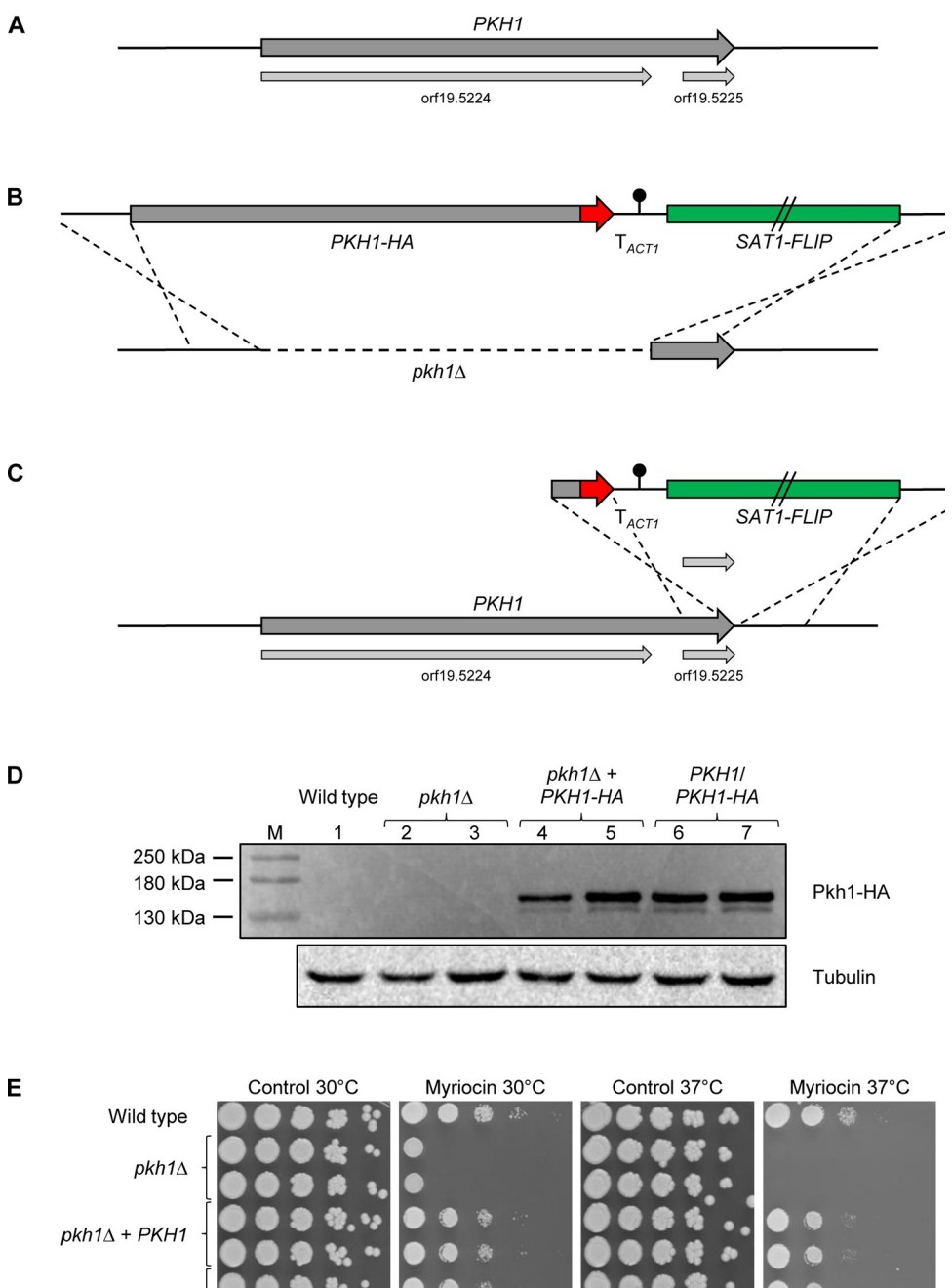

**Fig 5. *PKH1* comprises orf19.5224 and orf19.5225.** (A) Schematic of the *PKH1* locus in *C. albicans* strain SC5314. The *PKH1* coding sequence and the annotated orf19.5224 and orf19.5225 are represented by arrows. (B) Integration of an HA-tagged *PKH1* copy into *pkh1Δ* mutants. (C) Integration of a truncated version of the *PKH1-HA* cassette into the endogenous *PKH1* locus of the wild-type strain SC5314. (D) Detection of HA-tagged Pkh1 by Western blotting with an anti-HA antibody. Photos of the Western blot signals and the membrane were overlaid to show size markers (M). The untagged wild-type strain SC5314 and the parental *pkh1Δ* mutants served as negative controls. Reprobing with an anti-tubulin antibody was used to control for equal loading. (E) Overnight cultures of the wild-type strain SC5314, *pkh1Δ* mutants, and complemented strains were serially diluted and spotted on YPD agar plates containing 1 μg/ml myriocin or the solvent DMSO (control). Plates were incubated for 48 h at 30˚C or 37˚C. Results for both independently generated series of strains are shown in (D) and (E).

allele at the native locus in the *pkh1Δ* mutants (in which only the orf19.5224 part had been deleted, Fig 5B). Using an anti-HA antibody, a protein that was even larger than the expected size (132 kDa) was detected by Western blotting in the strains with the HA-tagged *PKH1* allele (Fig 5D, lanes 4 and 5). To exclude the presence of an undetected stop codon between orf19.5224 and orf19.5225 in the wild-type strain SC5314, we inserted a truncated cassette that contained only a part of the HA-tagged *PKH1*, starting at a PvuII site located 11 nucleotides in front of the orf19.5225 initiation codon, into either of the two endogenous loci (see Fig 5C). The same large protein was detected in the resulting strains (Fig 5D, lanes 6 and 7), confirming that orf19.5224 and orf19.5225 are not separate genes but part of *PKH1* (the predicted size of an HA-tagged protein encoded by orf19.5225 alone would be 16.9 kDa). Although the *pkh1Δ* mutants exhibited wild-type growth on YPD medium, they were hypersusceptible to myriocin (Fig 5E), in line with a function of Pkh1 in the Ypk1 signaling pathway. This phenotype was largely reverted after introduction of either a wild-type *PKH1* allele or the HA-tagged copy, demonstrating that the HA-tagged Pkh1 was functional.

The most likely candidate for a protein kinase that acts redundantly with Pkh1 in Ypk1 phosphorylation at the PDK1 site is Pkh3 (encoded by orf 19.1196), which has 68% similarity to Pkh1 in the N-terminal part containing the kinase domain. In *S. cerevisiae*, *PKH3* was isolated as a high-copy-number suppressor of the temperature-sensitive growth defect of a *pkh1*[D398G] *pkh2Δ* double mutant, but *PKH3* overexpression did not rescue the inviability of *pkh1Δ pkh2Δ* cells, demonstrating that Pkh3 cannot compensate for the loss of Pkh1 and Pkh2 in *S. cerevisiae* [5,29]. Deletion of *PKH3* in *C. albicans* strain SC5314 did not cause a growth defect and, unlike *pkh1Δ* mutants, *pkh3Δ* mutants were not hypersusceptible to myriocin, except for a slightly increased susceptibility to higher myriocin concentrations at 37°C (Fig 6A). Initial attempts to generate *pkh1Δ pkh3Δ* double mutants were unsuccessful, since we could replace only one of the two *PKH1* alleles, but not the remaining allele, in a *pkh3Δ* background. However, screening for slowly growing transformants after prolonged incubation on the selection plates enabled us to isolate mutants in which also the second *PKH1* allele in *pkh3Δ* mutants or the second *PKH3* allele in *pkh1Δ* mutants was deleted. Direct comparison showed that the growth defect of *pkh1Δ pkh3Δ* double mutants was even more severe than that of *ypk1Δ* mutants (Fig 6B), and reintroduction of a single copy of either *PKH1* or *PKH3* into the double mutants restored growth (Fig 6C). These observations indicated that (a) the combined functions of Pkh1 and Pkh3 are important for growth of *C. albicans* even under optimal conditions on rich medium, (b) Pkh1 and Pkh3 may act redundantly in Ypk1 phosphorylation, although Pkh1 is more important for Ypk1 activity when sphingolipid biosynthesis is inhibited (as indicated by the myriocin hypersensitivity of *pkh1Δ* but not *pkh3Δ* mutants), and (c) Pkh1 and Pkh3 have additional functions beyond Ypk1 activation (as indicated by the more severe growth defect of *pkh1Δ pkh3Δ* double mutants compared to *ypk1Δ* mutants), in line with the known function of Pkh1/2 in activation of the PKC1 pathway in *S. cerevisiae* [5].

To obtain direct evidence that both Pkh1 and Pkh3 can mediate Ypk1 phosphorylation, we detected T548-phosphorylated Ypk1 in the wild type and mutants lacking Pkh1 and/or Pkh3. For this purpose, we used an anti-P-PKC antibody, which recognizes the Pkh1/2-phosphorylated form of Ypk1 in *S. cerevisiae* [33]. In addition to Ypk1 and Ypk2, the protein kinases Pkc1 and Sch9 are also phosphorylated by Pkh1/2 in *S. cerevisiae* [24], and a control experiment that included *pkc1Δ*, *sch9Δ*, and *ypk1Δ* mutants enabled us to identify the bands that corresponded to Ypk1 on Western blots (Fig 7A). In *pkh1Δ* single mutants, the amount of T548-phosphorylated Ypk1 was decreased (average signal strength was 38% of wild-type levels), while no major change was observed in *pkh3Δ* single mutants (average signal strength was 92% of wild-type levels), but the Ypk1-specific signals disappeared in the *pkh1Δ pkh3Δ* double mutants (Fig 7B). This result demonstrates that Pkh1 and Pkh3 have a redundant function in Ypk1 phosphorylation in *C.*

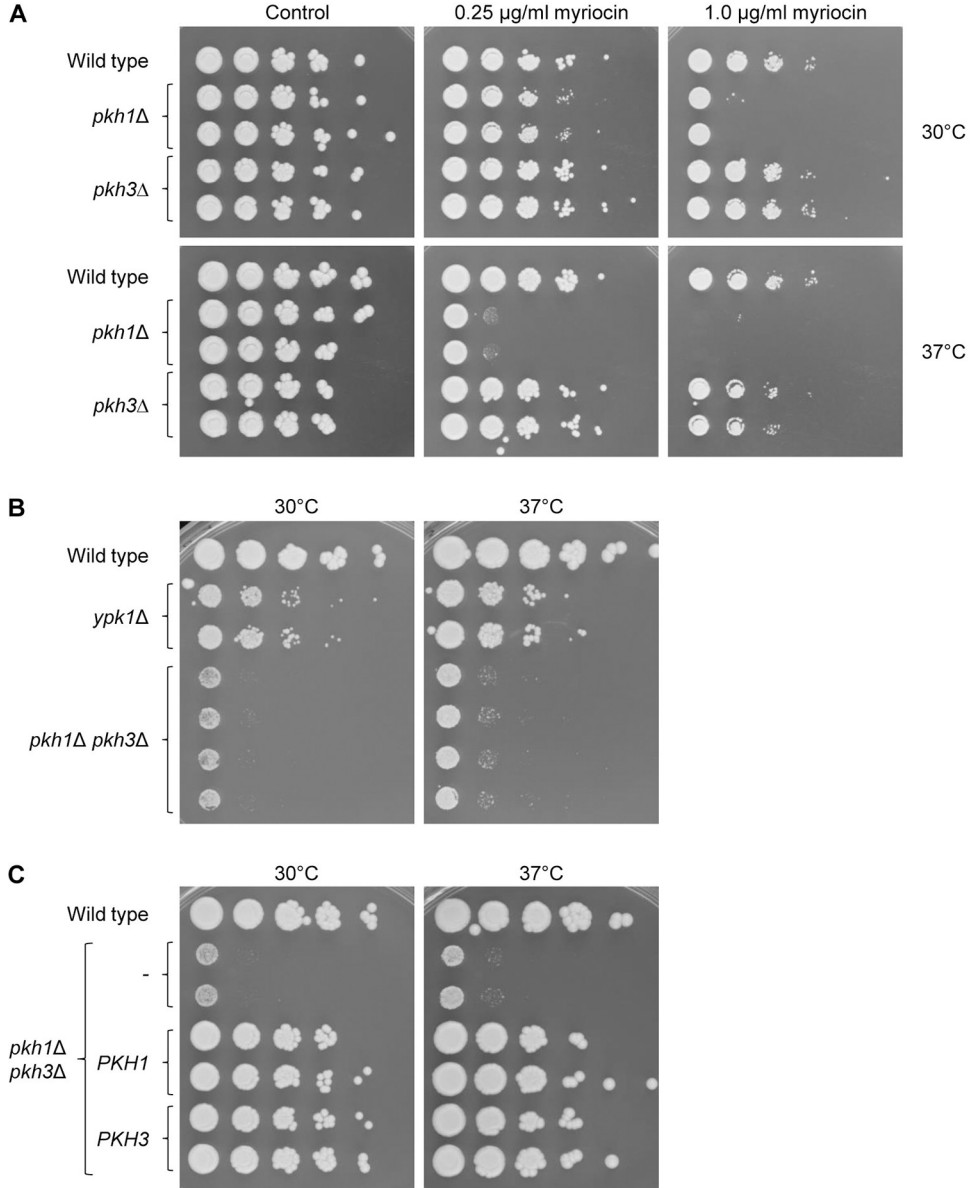

**Fig 6. Phenotypes of cells lacking Pkh1 and/or Pkh3.** (A) Overnight cultures of the wild-type strain SC5314, *pkh1Δ* mutants, and *pkh3Δ* mutants were serially diluted and spotted on YPD agar plates with the indicated myriocin concentrations. Control plates contained the solvent DMSO. (B) Comparison of the growth of *ypk1Δ* mutants and four independently generated *pkh1Δ pkh3Δ* double mutants on YPD medium. (C) Growth of the wild type, *pkh1Δ pkh3Δ* double mutants, and complemented strains in which a single copy of either *PKH1* or *PKH3* was reintroduced on YPD medium. Plates were incubated for 48 h at 30˚C or 37˚C in (A) to (C).

*albicans*. Interestingly, reduced amounts of phosphorylated Pkc1 (and Sch9) were still detectable in the *pkh1Δ* and *pkh1Δ pkh3Δ* mutants, indicating that phosphorylation of Pkc1 in its activation loop can be mediated by a protein kinase other than Pkh1 and Pkh3.

## The Ypk1 signaling pathway is important for hyphal growth of *C. albicans*

Proper membrane lipid composition is important for hyphal growth of *C. albicans*, an important virulence attribute of this pathogenic yeast [34–41]. Since the Ypk1 signaling pathway

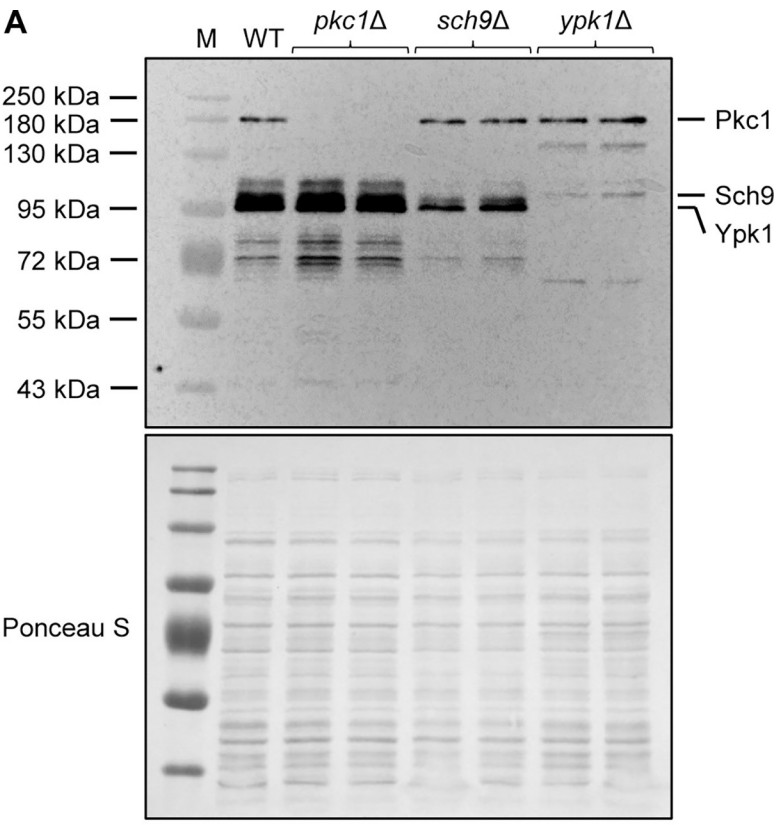

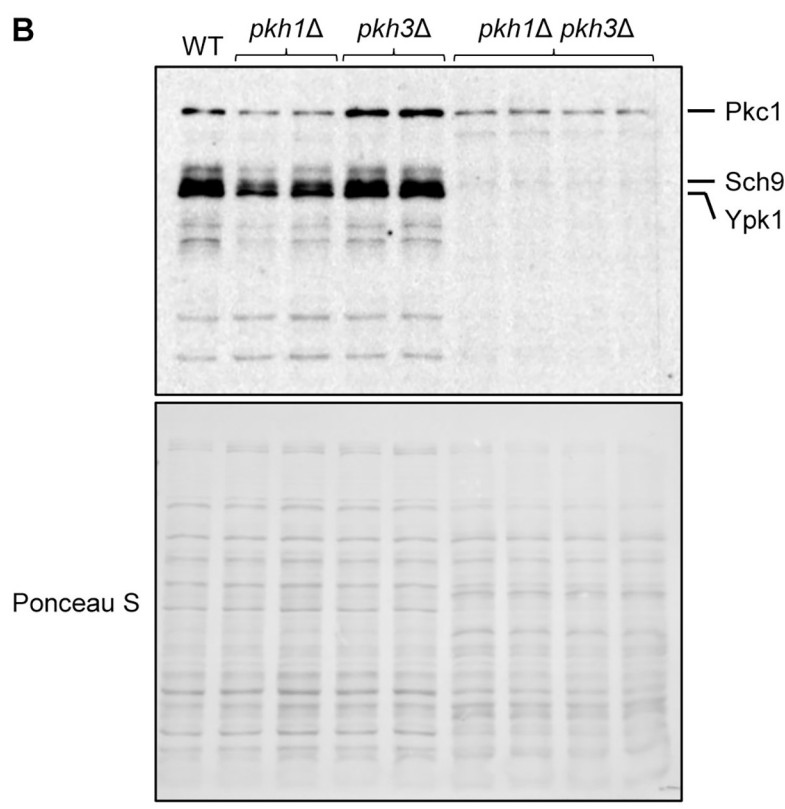

**Fig 7. Pkh1 and Pkh3 have a redundant role in Ypk1 phosphorylation.** (A) Detection of Pkc1, Sch9, and Ypk1 phosphorylated at their PDK sites in the wild-type strain SC5314 (WT) and *pkc1Δ*, *sch9Δ*, and *ypk1Δ* mutants by Western blotting with an anti-P-PKC1 antibody. Photos of the Western blot signals and the membrane were overlaid to show the size marker (M). (B) Detection of P-Ypk1 in wild-type cells, *pkh1Δ* and *pkh3Δ* single mutants, and *pkh1Δ pkh3Δ* double mutants. For each gene, independently generated mutants were used in (A) and (B). Ponceau S staining was used as a loading control.

controls membrane lipid homeostasis, we compared the ability of mutants lacking different components of the pathway to form hyphae in response to serum, a strong inducer of morphogenesis. As can be seen in Fig 8, the *ypk1Δ* mutants were unable to form true hyphae and either remained in the yeast form or produced short pseudohyphae under these conditions. The hyphal growth defect of cells lacking Ypk1 was not rescued by the additional deletion of *FPK1*, although pseudohyphae formed by the *ypk1Δ fpk1Δ* double mutants were not as thick as those of the *ypk1Δ* mutants. The *pkh1Δ* and *pkh3Δ* single mutants formed normal hyphae, but

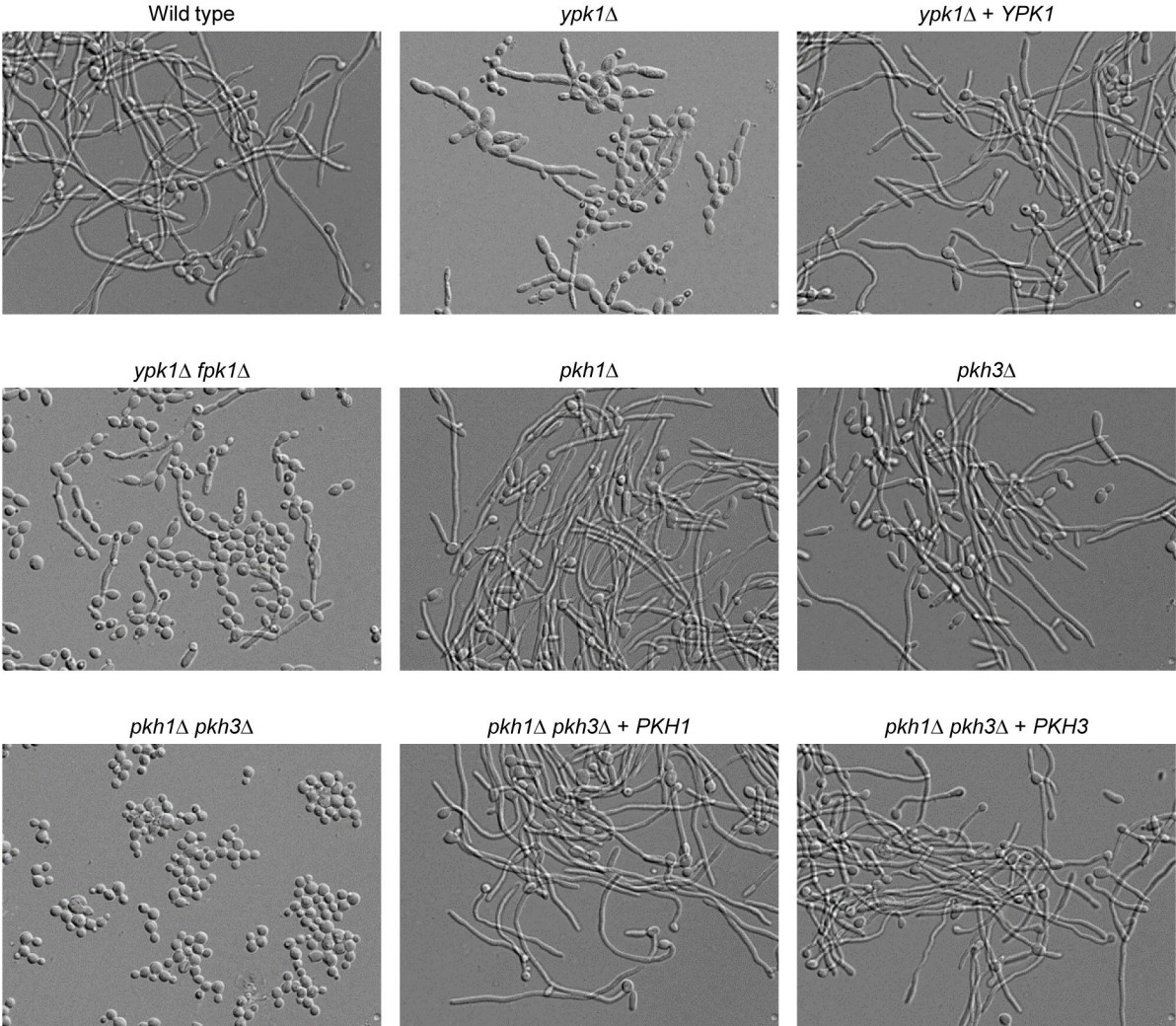

**Fig 8. The Ypk1 signaling pathway is important for hyphal growth of *C. albicans*.** YPD overnight cultures of the indicated strains were diluted 1:100 in YPD + 10% serum and incubated for 5 h at 37°C. Independently constructed mutants behaved identically and only one of them is shown in each case.

deletion of both *PKH1* and *PKH3* completely abolished filamentous growth. Reintroduction of either of these genes into the *pkh1Δ pkh3Δ* double mutants restored the ability to form hyphae, in line with the redundant function of Pkh1 and Pkh3 in Ypk1 activation. The more severe filamentation defect of the *pkh1Δ pkh3Δ* double mutants compared to *ypk1Δ* mutants is consistent with their stronger general growth defect (Fig 6B) and presumably additional functions beyond Ypk1 activation.

## Discussion

Elucidating the roles of the many protein kinases that are involved in the signaling networks of cells is important to understand how organisms regulate their activities and how they adapt to changes in their environment. There is also strong interest in knowing which genes are essential for viability in pathogenic microbes such as *C. albicans*, as the encoded proteins are considered as ideal potential targets for the development of novel antiinfective drugs [4,42,43].

In the case of *C. albicans*, failure to generate homozygous deletion or insertion mutants is usually a first indication that a gene might be essential. In the *Candida* Genome Database, genes are listed as putative essential when no homozygous mutants were obtained by the *UAU* method, which selects for two copies of the insertion marker and results in triplication and retention of a functional copy of essential target genes [44]. The isolation of haploid *C. albicans* strains also enabled the generation of a genome-wide comprehensive set of transposon insertion mutants and the identification of 1,195 high-confidence essential genes [4]. Both of these approaches provided evidence that *YPK1* is an essential gene, as is the case for its orthologs *YPK1/2* in *S. cerevisiae* [7]. In contrast, a recent study that used a repressible promoter to predict gene essentiality for a large library of conditional mutants concluded that *YPK1* is not essential [42]. However, growth of knock-down mutants might also be due to insufficient repression of the target gene. In our present study, we used inducible gene deletion to obtain true *ypk1Δ* null mutants and study their terminal phenotype before cell death, but unexpectedly found that these mutants are viable and can grow, albeit slowly. With the same approach we had recently demonstrated that mutants lacking Snf1, another kinase that was thought to be essential in *C. albicans*, are viable and can grow under optimal conditions [22]. The work presented here again demonstrates that FLP-mediated induced gene deletion is a powerful approach to obtain definite proof of whether a gene is essential or not in *C. albicans*.

*C. albicans* is not the only pathogenic fungus in which Ypk1 is not essential. The basidiomyceteous yeast *Cryptococcus neoformans* possesses single *YPK1* and also *PKH1* (*PDK1*) orthologs, which are not essential for viability in this organism [45]. *C. neoformans pdk1Δ* mutants exhibited strongly reduced growth on YPD plates, whereas *ypk1Δ* mutants showed a milder growth defect. A Δ*ypkA* mutant of the pathogenic mold *Aspergillus fumigatus* had a very severe growth defect but was viable [46]. In contrast, in its relative *Aspergillus nidulans*, the single *YPK1* (*ypkA*) and *PKH1* (*pkhA*) homologs are essential for viability [47]. Therefore, whether the Ypk1 signaling pathway is essential for viability varies among different fungi. It should also be stressed that the importance of specific genes for certain phenotypes, including viability, depends on the genetic background and may even differ between strains of the same species. While in *S. cerevisiae* Pkh1/2 were found to be essential in most studies, *pkh1Δ pkh2Δ* double mutants described in one report grew slowly at 24˚C and growth at 30˚C was partially restored by sorbitol [26].

The phenotypes of the *ypk1Δ* mutants provided evidence that the well established role of Ypk1 orthologs in regulating sphingolipid biosynthesis and cell membrane lipid asymmetry is conserved in *C. albicans*. We did not make efforts to measure serine-palmitoyl transferase and ceramide synthase enzyme activities or determine the lipid composition of the cell membrane,

but the hypersensitivity of the *ypk1Δ* mutants to the sphingolipid biosynthesis inhibitor myriocin, the observed negative regulation of the downstream kinase Fpk1, and the partial rescue of *ypk1Δ* mutant phenotypes by *FPK1* deletion support this function. Interestingly, the *ypk1Δ* mutants did not exhibit an obvious endocytosis defect, as assessed by FM4-64 internalization, and could also take up exogenously supplied fatty acids, suggesting that some phenotypic consequences of *YPK1* deletion are milder than in *S. cerevisiae*, which may explain why *C. albicans ypk1Δ* mutants are viable. A surprising finding that has already been pointed out by other researchers [31] was that *C. albicans* mutants lacking the Ypk1-activating kinase Pkh1 grow normally under unstressed conditions. We first hypothesized that phosphorylation of Ypk1 at the PDK1 site in its activation loop might be less critical for Ypk1 function in *C. albicans*, but this turned out not to be the case, since cells containing the *YPK1*[T548A] allele grew as poorly as *ypk1Δ* mutants. In contrast, we found that Pkh1 is dispensable because Pkh3, a kinase that to our knowledge has not been implicated in the Ypk1 signaling pathway, can substitute Pkh1 in *C. albicans*.

So far, little was known about the function of Pkh3. Konstantinidou and Morrissey reported that *pkh3* transposon insertion mutants had defects in filamentation and biofilm formation [48]. However, no such defects were observed in earlier studies using the same or independently constructed *pkh3Δ* mutants [3,32]. In accordance with the latter studies, we also did not see a filamentation defect in our *pkh3Δ* mutants generated from the wild-type strain SC5314 when hyphal growth was induced by serum (Fig 8). Douglas *et al.* found that the sensitivity of a *pkh1Δ* mutant to cell wall (Congo red) and cell membrane (SDS) stress was exacerbated by additional deletion of one of the *PKH3* alleles, and they were unable to generate homozygous *pkh1Δ pkh3Δ* double mutants, indicating that the combined functions of these kinases are important for cellular fitness and possibly viability [31]. A follow-up study by Wang *et al.* assigned Pkh1 and Pkh3 a role in the regulation of membrane furrows known as eisosomes [49], but so far no function of Pkh3 in the Ypk1 signaling pathway was recognized. While the enhancement of growth defects upon inactivation of two different protein kinases as compared with those of single mutants does not necessarily implicate them in the same signaling pathway, we have provided direct proof that Pkh1 and Pkh3 have a redundant function in Ypk1 activation, by showing that both can mediate phosphorylation of Ypk1 at the PDK1 site in the activation loop (Fig 7). Under nonstressed conditions, Pkh3 can sufficiently substitute Pkh1 for Ypk1 activation (and possibly other functions) to allow normal growth, explaining the puzzling observation made in this and previous studies that *C. albicans pkh1* mutants do not exhibit obvious growth defects [3,31,32]. In summary, our study demonstrates that, despite its conserved important function in membrane lipid homeostasis, Ypk1 is not essential for viability in *C. albicans*, as was previously thought, and that the Ypk1 signaling pathway in this pathogenic yeast includes Pkh3 as an additional upstream kinase for Ypk1 activation (Fig 9).

## Materials and methods

### Strains and growth conditions

The *C. albicans* strains used in this study are listed in S1 Table. All strains were stored as frozen stocks with 17.2% glycerol at -80°C and subcultured on YPD agar plates (10 g yeast extract, 20 g peptone, 20 g glucose, 15 g agar per liter) at 30°C. Strains were routinely grown in YPD liquid medium at 30°C in a shaking incubator. An exception were the slowly growing *ypk1Δ* mutants and *phk1Δ pkh3Δ* double mutants, which were subcultured at 37°C, at which they grew somewhat better than at 30°C. For the selection of transformants, 200 μg/ml nourseothricin (Werner Bioagents, Jena, Germany) or 1 mg/ml hygromycin B was added to YPD agar plates. To obtain nourseothricin-sensitive derivatives in which the *SAT1* flipper cassette was excised by

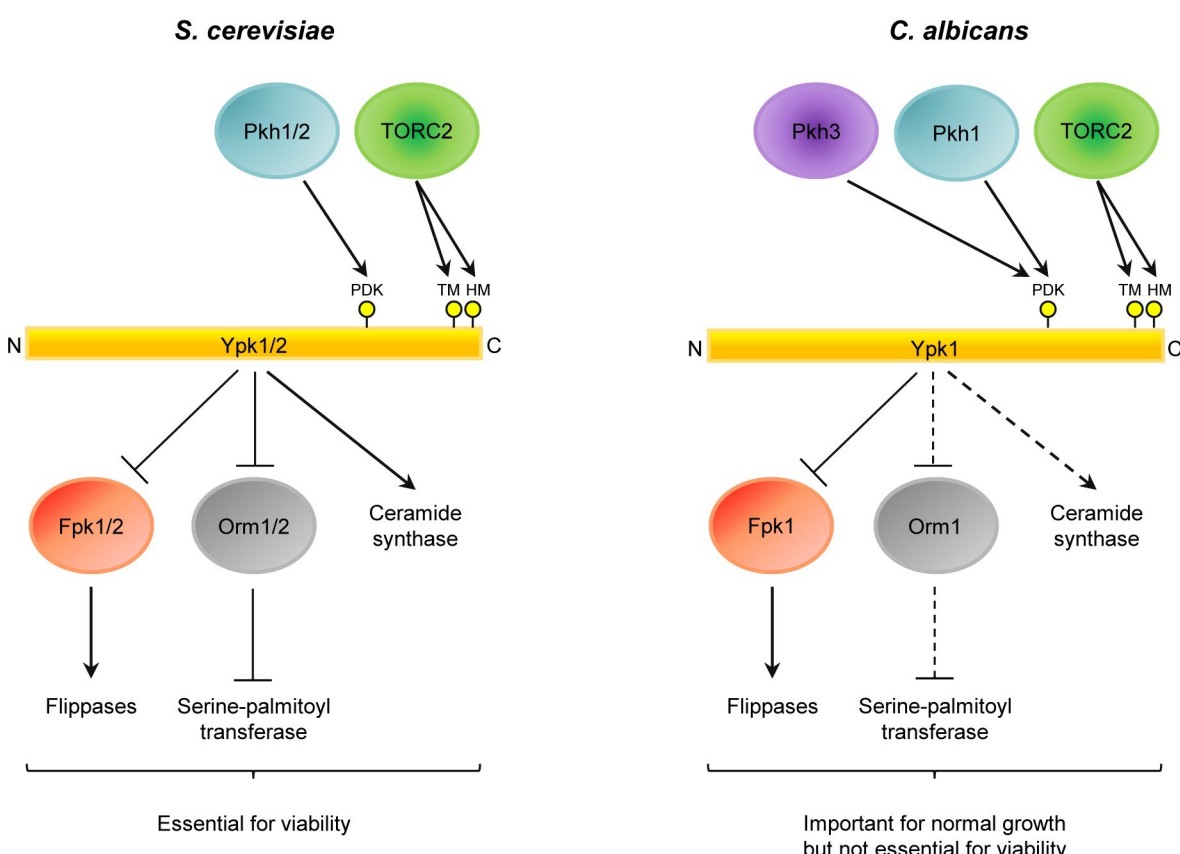

**Fig 9. Comparison of the Ypk1 signaling pathways in *S. cerevisiae* and *C. albicans*.** *S. cerevisiae* contains two paralogs of the protein kinases Pkh1/2, Ypk1/2, and Fpk1/2, and of the serine-palmitoyltransferase inhibitors Orm1/2, whereas *C. albicans* possesses only one ortholog of each of these. Ypk1 is represented by a linear bar to illustrate key phosphorylation sites; the other kinases and Orm proteins are shown as ovals. Arrows indicate activation, end lines inhibition. The importance of the phosphorylation sites in the activation loop (PDK), turn motif (TM), and hydrophobic motif (HM) for Ypk1 function is conserved, and our results indicate that Ypk1 regulates membrane asymmetry by inhibiting Fpk1 and thereby downregulating the activity of flippases also in *C. albicans*. As indicated by the stippled lines, the assumed role of Ypk1 in sphingolipid synthesis still has to be experimentally addressed in *C. albicans*. The major difference in the Ypk1 signaling pathways of the two species is that *C. albicans* has integrated a novel upstream kinase, Pkh3, that acts redundantly with Pkh1 to phosphorylate and thereby activate Ypk1 at the PDK1 site, and that sufficient membrane functionality is retained in the absence of Ypk1 to retain cell viability and allow slow growth.

FLP-mediated recombination, transformants were grown overnight in YCB-BSA-YE medium (23.4 g yeast carbon base, 4 g bovine serum albumin, 2 g yeast extract per liter, pH 4.0) without selective pressure to induce the *SAP2* promoter controlling *caFLP* expression. Appropriate dilutions were plated on YPD agar plates and grown for 2 days at 30°C. Individual colonies were picked and streaked on YPD plates as well as on YPD plates with 100 μg/ml nourseothricin to confirm sensitivity. To induce hyphal growth, YPD overnight cultures of the strains were diluted 1:100 in 10 ml YPD medium containing 10% heat-inactivated fetal bovine serum (Thermo Fisher Scientific) and incubated for 5 h at 37°C. The cells were fixed for 1 h with 4% formaldehyde and imaged with a Leica DMI6000 microscope.

## Strain constructions

*C. albicans* strains were transformed by electroporation as described previously [50]. To generate a FLP-deletable *YPK1* cassette, the *YPK1* (orf19.399) coding region and 1515 bp of upstream and 539 bp of downstram sequences was amplified from genomic DNA of strain

SC5314 with primers YPK1ex3-1 and YPK1ex3-2 (S2 Table) and substituted for the *SNF1* gene in the previously described pSNF1ex3 [22], yielding pYPK1ex3. The cassette was excised from the vector backbone and used to transform the heterozygous *YPK1/ypk1Δ* mutants SCYPK1M2A and -B [1], resulting in strains SCYPK1M3A and -B. The previously described *YPK1* deletion cassette from plasmid pYPK1M1 [1] was then used to delete the second endogenous *YPK1* allele in these strains to obtain SCYPK1M4A and -B. These strains were grown overnight in YCB-BSA medium to induce *caFLP* expression. This resulted in the derivatives SCYPK1M5A and -B, in which both the *SAT1* flipper cassette and the FLP-deletable ectopic *YPK1* copy were deleted, and strains SCYPK1M6A and -B, which had excised only the *SAT1* flipper cassette but retained the ectopic *YPK1* copy. The insert from plasmid pSAP2FL1 [22] was used to integrate the *ecaFLP* gene under control of the *SAP2-1* promoter into the *SAP2-1* allele of strains SCYPK1M6A and -B to generate the conditional *ypk1Δ* mutants SCYPK1M7A and -B. Strains SCYPK1M8A and -B were obtained after excision of the FLP-deletable *YPK1* copy from strains SCYPK1M7A and -B. The *YPK1* deletion cassette from plasmid pYPK1M1 was also used to delete the second endogenous *YPK1* allele in the heterozygous mutants SCYPK1M2A and -B, followed by excision of the *SAT1* flipper cassette to obtain the homozygous *ypk1Δ* mutants SCYPK1M22A and -B. For reintroduction of a functional *YPK1* copy, the *YPK1* coding sequence plus flanking upstream and downstream sequences was amplified with primers YPK1.01 and YPK1.05 and the PCR product substituted for the 5'*YPK1* fragment in pYPK1M1 to obtain pYPK1K1. The insert from this plasmid was integrated at the original locus of the *ypk1Δ* mutants SCYPK1M22A and -B, followed by recycling of the *SAT1* flipper cassette to generate the complemented strains SCYPK1MK2A and -B. To obtain *ypk1Δ fpk1Δ* double mutants, the insert from pYPK1M1 was used to sequentially delete the *YPK1* alleles in the *fpk1Δ* mutants SC223M4A and -B [2], yielding strains SCΔ*fpk1*YPK1M3A and -B. The *SAT1*-Flipper cassette was not excised from the double mutants, because we observed that they produced heterogeneous progeny (likely suppressor mutants) after further passaging in YCB-BSA medium.

To construct 3xHA-tagged *YPK1* alleles, the *YPK1* coding region starting from position +18 was amplified with primers YPK1.11 and YPK1.10; the latter primer introduces a KasI site, encoding a Gly-Ala linker, instead of the *YPK1* stop codon. A fragment from plasmid pCEK1H1 [51], encoding three copies of the HA epitope followed by a stop codon and the *ACT1* transcription termination sequence, was amplified with primers HA-KasI and ACT19. The two PCR products were digested with SacI/KasI and KasI/SacII, respectively, and substituted for the 5'*YPK1* fragment in the SacI/SacII-digested pYPK1M1 to obtain pYPK1H1. The 3xHA-tagged *YPK1*$^{T548A}$ allele was generated by a fusion PCR with the primer pairs YPK1.11/YPK1T548A-R and YPK1T548A-F/YPK1.10 using pYPK1H1 as template; the overlapping primers YPK1T548A-R and YPK1T548A-F change the threonine codon ACA (positions +1642 to +1644 in *YPK1*) into the alanine codon GCA. The PCR product was digested with SacI/KasI and substituted for the wild-type *YPK1* sequence in pYPK1H1 to obtain pYPK1$^{T548A}$H1. The 3xHA-tagged *YPK1*$^{S687A}$ and *YPK1*$^{T705A}$ alleles were generated in an analogous fashion using the primer pairs YPK1.11/YPK1T687A-R and YPK1T687A-F/ACT19, and YPK1.11/YPK1T705A-R and YPK1T705A-F/ACT19, respectively, resulting in plasmids pYPK1$^{S687A}$H1 and pYPK1$^{T705A}$H1. The inserts from these plasmids were used to replace the remaining wild-type *YPK1* allele in the heterozygous mutants SCYPK1M2A and -B, followed by recycling of the *SAT1* flipper cassette, to obtain strains SCYPK1H2A and -B, SCYPK1$^{T548A}$H2A and -B, SCYPK1$^{S687A}$H2A and -B, and SCYPK1$^{T705A}$H2A and -B. A 3xMyc-tagged *YPK1* was obtained by amplifying a part of the *YPK1* coding region, starting from position +1471, with primers YPK1.08 and YPK1.09; the latter primer introduces a BamHI site, encoding a Gly-Ser linker, instead of the *YPK1* stop codon. The PCR product was

digested with SacI and BamHI and inserted together with a BamHI-SacII fragment from pKIS1Myc3x [52], encoding three copies of the Myc epitope followed by a stop codon and the *ACT1* transcription termination sequence, instead of the 5'*YPK1* fragment in the SacI/SacII-digested pYPK1M1. The insert from the resulting plasmid pYPK1Myc3 was used to replace one of the wild-type *YPK1* alleles in the wild-type strain SC5314 and the *fpk1*Δ mutants, followed by recycling of the *SAT1* flipper cassette, to obtain strains SCYPK1Myc32A and -B and SCΔ*fpk1*YPK1Myc32A and -B, respectively.

To construct a 3xHA-tagged *FPK1*, the *FPK1* (orf19.223) upstream and coding sequence was amplified with primers 223.05 and 223.06; the latter primer introduces a BamHI site, encoding a Gly-Ser linker, instead of the *FPK1* stop codon. The PCR product was digested with SacI and BamHI and inserted together with a BamHI-SacII fragment from pMIG1H1 [52], encoding three copies of the HA epitope followed by a stop codon and the *ACT1* transcription termination sequence, instead of the upstream flanking sequence of the *FPK1* deletion cassette in the SacI/SacII-digested plasmid p223M1 [2]. The insert from the resulting plasmid p223H1 was integrated at the original locus (allele 1) in the *fpk1*Δ mutants SC223M4A and -B [2], followed by excision of the *SAT1* flipper cassette to obtain strains SC223MH2A and -B. The HA-tagged *FPK1* was integrated in the same way in the wild-type strain SC5314 to generate strains SC223H2A and -B. Since the homozygous *ypk1*Δ mutants were difficult to transform, the HA-tagged *FPK1* was first integrated in the heterozygous *YPK1/ypk1*Δ mutants SCYPK1M2A and -B, followed by deletion of the second wild-type *YPK1* allele to generate SCYPK1M4FPK1H2A and -B.

For reintroduction of a functional *PKH1* copy into *pkh1*Δ mutants, the *PKH1* (orf19.5224-orf19.5225) coding sequence plus flanking upstream and downstream sequences was amplified with primers PKH1.01 and PKH1.11 and the PCR product substituted for the 5'*PKH1* fragment in the *PKH1* deletion cassette of plasmid pPKH1M1 [1] to obtain pPKH1K2. Sequencing showed that pPKH1K2 contained *PKH1* allele 1. The insert from this plasmid was integrated in one of the *pkh1*Δ alleles of mutants SCPKH1M4A and -B [1], followed by recycling of the *SAT1* flipper cassette to generate the complemented strains SCPKH1MK2A and -B. A 3xHA-tagged *PKH1* was generated by amplifying the *PKH1* coding and upstream sequences with primers PKH1.01 and PKH1.12; the latter primer introduces a KasI site instead of the *PKH1* stop codon. The PCR product was digested with SacI/KasI and inserted together with the KasI-SacII fragment from pYPK1H1, containing the 3xHA and *ACT1* transcription termination sequences, instead of the upstream flanking sequence of pPKH1M1 to yield pPKH1H1. Sequencing showed that pPKH1H1 contained *PKH1* allele 2. The insert from this plasmid was integrated in one of the *pkh1*Δ alleles of mutants SCPKH1M4A and -B (see Fig 5B), followed by recycling of the *SAT1* flipper cassette to generate strains SCPKH1M4H12A and -B. pPKH1H3 is a derivative of pPKH1H1 in which the correct *PKH1* downstream sequence, amplified with primers PKH1.15 and PKH1.16, was inserted instead of the orf19.5224 downstream sequence and *PKH1* allele 2 was replaced by allele 1, which contains a unique PvuII site between orf19.5224 and orf19.5225. A PvuII-ApaI fragment from pPKH1H3 was used to replace one of the endogenous *PKH1* alleles of the wild-type strain SC5314 by the HA-tagged *PKH1* (see Fig 5C), followed by recycling of the *SAT1* flipper cassette to generate strains SCPKH1H32A and -B. *pkh1*Δ *pkh3*Δ double mutants were generated by sequential deletion of the *PKH1* alleles in the *pkh3*Δ mutants SCPKH3M4A and -B with the insert from pPKH1M1, yielding strains SCΔ*pkh3*PKH1M4A and -B, and also by sequential deletion of the *PKH3* alleles in the *pkh1*Δ mutants SCPKH1M4A and -B with the insert from pPKH3M1 [1], yielding strains SCΔ*pkh1*PKH3M4A and -B. Single copies of *PKH1* or *PKH3* were reintroduced into the double mutants using the inserts from plasmids pPKH1K2 and pPKH3K1 (described below), generating the complemented strains

SCΔ*pkh1*Δ*pkh3*PKH1MK2A, SCΔ*pkh1*Δ*pkh3*PKH3MK2B, SCΔ*pkh3*Δ*pkh1*PKH1MK2A and SCΔ*pkh3*Δ*pkh1*PKH3MK2B. pPKH3K1 was generated by amplifying the *PKH3* coding sequence plus flanking upstream and downstream sequences with primers PKH3.05 and PKH3.06 and substituting the PCR product for the 5'*PKH3* fragment in the *PKH3* deletion cassette of plasmid pPKH3M1. The correct genomic integration of all constructs and subsequent excision of the *SAT1* flipper cassette were confirmed by Southern hybridization using the flanking sequences as probes.

## Isolation of genomic DNA and Southern hybridization

Genomic DNA from *C. albicans* strains was isolated as described previously [53]. The DNA was digested with appropriate restriction enzymes, separated on a 1% agarose gel, transferred by vacuum blotting onto a nylon membrane, and fixed by UV crosslinking. Southern hybridization with enhanced chemiluminescence-labeled probes was performed with the Amersham ECL Direct Nucleic Acid Labelling and Detection System (Cytiva) according to the instructions of the manufacturer.

## Growth assays

For the dilution spot assays, YPD overnight cultures of the strains were adjusted to an optical density at 600 nm ($OD_{600}$) of 2.0 in water, serially 10-fold diluted, and spotted on YPD plates with or without inhibitors and supplements (all from Merck) as explained in the legends to the figures. Myriocin and cerulenin were dissolved in DMSO, and fatty acids in Brij58. Control plates contained corresponding concentrations of the solvents (DMSO, Brij58, or both DMSO and Brij58; the latter two are not included in Fig 2B since there was no difference to the DMSO plates). Plates were incubated for 2 days at 30°C or 37°C. The disk diffusion assay was performed as described in the legend to S2 Fig. To obtain growth curves, individual colonies of the strains were picked from YPD plates and adjusted to an $OD_{600}$ of 0.05 in 200 μl YPD in a sterile 96-well plate. The strains were grown at 37°C in an Infinite M Plex Reader (Tecan Group Ltd., Switzerland) with shaking at an orbital amplitude of 4 mm and the $OD_{600}$ was read every 10 min for 23 h. Three biological replicates, each with three technical replicates, were performed per strain. Doubling times were estimated using the R package Growthcurver [54].

## Vacuole staining

Overnight cultures of the strains were diluted to an $OD_{600}$ of 0.4 in 5 ml YPD. After 2 h of incubation at 30°C, two 1-ml aliquots of each culture were centrifuged and the cells resuspended in 50 μl of cold YP + 40 μM FM 4–64 (AAT Bioquest Inc). After 20 min of incubation at 4°C, the cells were washed with cold YPD and resuspended in 1 ml YPD. After 15 and 30 min of incubation at 30°C, the cells were washed two times with cold YNB and kept at 4°C before being imaged with a fluorescence microscope using appropriate filters for red fluorescence detection.

## Actin staining

Overnight cultures of the strains were diluted to an $OD_{600}$ of 0.2 in 10 ml YPD. After 3 h of incubation at 30°C, formaldehyde was added to a final concentration of 4.4% and the cultures were incubated for 45 min at room temperature. The cells were washed three times with PBS and stained with Alexa Fluor 488-conjugated Phalloidin (Lonza) overnight at 4°C. Images were captured with a Leica DMI6000 fluorescence microscope.

## Western blotting

Overnight cultures of the strains were diluted to an $OD_{600}$ of 0.4 in fresh YPD and grown for 5 h at 30°C. Cells were collected by centrifugation, washed with ice-cold water, and resuspended in 300 μl breaking buffer (50 mM Tris-HCl pH 8, 250 mM NaCl, 5 mM EDTA, 0.1% [v/v] Triton X-100, cOmplete EDTA-free Protease Inhibitor Cocktail and PhosStop Phosphatase Inhibitor Cocktail [Roche]). An equal volume of 0.5 mm acid-washed glass beads was added to each tube. Cells were mechanically disrupted on a FastPrep-24 cell-homogenizer (MP Biomedicals) with three 40 s pulses, with 5 min on ice between each pulse. Cell lysates were centrifuged at 21,000 x *g* for 15 min at 4°C, the supernatant was collected, and the protein concentration was quantified using the Bradford protein assay. Equal amounts of protein of each sample were mixed with one volume of 2 x Laemmli buffer, heated for 5 min at 95°C, and separated on an SDS-8% polyacrylamide gel. For the experiment shown in Fig 3C, cell extracts were separatd on $Zn^{2+}$-Phos-tag gels (25 μM Phos-tag, Wako Pure Chemical Industries) and electrophoresis was performed using Tris-MOPS-SDS buffer (0.1 M Tris base, 0.1 M MOPS, 0.1% SDS, 0.05 M sodium bisulfite) at 120 V for 165 min. Before protein transfer, the phos-tag gels were treated for 20 min in transfer buffer containing 10 mM EDTA. Separated proteins were transferred onto a nitrocellulose membrane with a mini-Protean System (Bio-Rad). To detect HA-tagged proteins, membranes were blocked with 5% milk in TBST and incubated overnight with rat monoclonal anti-HA-peroxidase antibody, clone 3F10 (Roche). For the detection of tubulin, membranes were blocked with 5% milk in TBST and incubated overnight at 4°C with rat anti-tubulin alpha antibody MCA 78G (Bio-Rad), washed with TBST, and then incubated with rabbit anti-rat HRP-conjugated antibody STAR21B (Bio-Rad). Myc-tagged Ypk1 was detected with anti-Myc (71D10) rabbit mAb (Cell Signaling Technology) and anti-rabbit HRP G-21234 (Invitrogen) as secondary antibody. T548-phosphorylated Ypk1 was detected with Phospho-PKC (pan) (zeta Thr410) (190D10) rabbit mAb (Cell Signaling Technology) and anti-rabbit HRP G-21234 as secondary antibody. To reprobe the immunoblots, membranes were incubated in stripping buffer (0.2 M glycine, 0.1% SDS, 1% Tween 20, pH 2.2), and washed in PBS and TBST before blocking with 5% milk. Signals were generated with the ECL chemiluminescence detection system (Cytiva) and captured with the ImageQuant LAS 4000 imaging system (Cytiva). Relative band intensities were calculated using Fiji software [55].

## Supporting information

**S1 Fig. Confirmation of the myriocin hypersensitivity of *ypk1Δ* mutants by a disk diffusion assay.**
(PDF)

**S2 Fig. Growth curves of the wild-type strain SC5314, *ypk1Δ* single mutants, and *ypk1Δ fpk1Δ* double mutants at 37°C.**
(PDF)

**S3 Fig. Comparison of the coding sequences of the cloned *PKH1-1* and *PKH1-2* alleles with the corresponding sequences in assembly 22 of the *C. albicans* genome sequence.**
(DOCX)

**S1 Table. *C. albicans* strains used in this study.**
(XLSX)

**S2 Table. Oligonucleotide primers used in this study.**
(XLSX)

## Author Contributions

**Conceptualization:** Bernardo Ramírez-Zavala, Joachim Morschhäuser.

**Data curation:** Bernardo Ramírez-Zavala, Joachim Morschhäuser.

**Formal analysis:** Bernardo Ramírez-Zavala, Joachim Morschhäuser.

**Funding acquisition:** Joachim Morschhäuser.

**Investigation:** Bernardo Ramírez-Zavala, Ines Krüger, Andreas Wollner, Sonja Schwanfelder.

**Methodology:** Bernardo Ramírez-Zavala, Ines Krüger.

**Project administration:** Joachim Morschhäuser.

**Resources:** Joachim Morschhäuser.

**Supervision:** Bernardo Ramírez-Zavala, Ines Krüger, Joachim Morschhäuser.

**Validation:** Bernardo Ramírez-Zavala, Joachim Morschhäuser.

**Visualization:** Bernardo Ramírez-Zavala, Joachim Morschhäuser.

**Writing – original draft:** Joachim Morschhäuser.

**Writing – review & editing:** Joachim Morschhäuser.

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
