## [Decision Letter · Decision Letter 0]

7 Jun 2023

Dear Dr Morschhauser,

Thank you very much for submitting your Research Article entitled 'The conserved Ypk1 protein kinase signaling pathway is rewired and not essential for viability in Candida albicans' to PLOS Genetics.

The manuscript was fully evaluated at the editorial level and by independent peer reviewers. The reviewers appreciated the attention to an important problem, but raised some concerns about the current manuscript. In particular, you should include some quantification of data presented, address a lack of focus in the writing, and consider adding a summary figure. Based on the reviews, we will not be able to accept this version of the manuscript, but we would be willing to review a revised version. 

If you decide to revise the manuscript for further consideration at PLOS Genetics, please aim to resubmit within the next 60 days, unless it will take extra time to address the concerns of the reviewers, in which case we would appreciate an expected resubmission date by email to plosgenetics@plos.org.

We are sorry that we cannot be more positive about your manuscript at this stage. Please do not hesitate to contact us if you have any concerns or questions.

Yours sincerely,

Anna Selmecki, Ph.D.

Academic Editor

PLOS Genetics

Geraldine Butler

Section Editor

PLOS Genetics

Reviewer's Responses to Questions

**Comments to the Authors:**

Reviewer #1: This study dissects the signaling pathway components and outputs associated with Candida albicans Ypk1, a highly conserved protein kinase that governs membrane structure. The study looks both upstream and downstream of Ypk1, relying upon guidance from the Ypk1 pathway in S. cerevisiae and other eukaryotes along with some judgment calls based on homology. Major conclusions are based upon deletion mutants, site-directed mutants, analysis of protein phosphorylation on Western blots, and mutant biological phenotypes. Major novel findings include:

1. YPK1 is not essential in C. albicans, in contrast with results from (at least) two previous independent studies. The game-changer here was application of the inducible FLP-deletion strategy, which the authors pioneered and used recently to show that another previously deduced essential gene (SNF1) is not essential.

2. Upstream kinase Pkh1 has overlapping function with Pkh3, a protein kinase with no known role in Ypk1 signaling in S. cerevisiae.

3. PKH1 has a longer coding region than deduced from assembly 22; it is actually a fusion of the ORFs 19.5224 and 19.5225. (The authors' finding is consistent with the Bruno RNASeq tracks displayed at CGD.)

The manuscript itself presents the work efficiently and thoughtfully. Every now and then I found myself thinking "Oh, they might want to consider this," and then a page later I'd read the authors' thoughts about just that consideration.

One really interesting observation is that this pathway has a role in hypha formation, as presented in Figure 8. The authors use an extremely strong inducing condition, hence the defects seem to be severe. The pkh1 pkh3 mutant in particular seems to grow exclusively as yeast cells. This might be an interesting finding to pursue in future studies.

One question I had as I read the manuscript was why YPK1 is not essential in C. albicans. I would be curious to hear the authors' thoughts. Maybe the authors could speculate about this issue in a sentence or two.

My only important criticism is that the manuscript does not have as clear a focal point as it could. Part of this is a consequence of the breadth of analysis that is presented. The one suggestion that I have in this regard is to create a summary figure that compares and contrasts the Ypk1 signaling pathway in S. cerevisiae and C. albicans. Either a pair of wiring diagrams or a simple table might provide the reader with a digestible reminder of the major differences and similarities.

One minor criticism is that the title seems to include an internal contradiction - that the pathway is conserved yet rewired. The authors might consider deleting the term "conserved."

Reviewer #2: This manuscript describes the phenotypes for C. albicans mutants lacking members of the conserved protein kinase cascade (Pkh, Ypk1, Fpk1). This kinase cascade was identified in S. cerevisiae as being important for regulating aspects of membrane function and stress resistance, including sphingolipid synthesis, endocytosis, and phospholipid asymmetry. A strong point of the paper is that they used a regulatable deletion system to demonstrate that the ypk1D mutant is very slow growing (whereas it had been assumed YPK1 was essential because it could not be mutated in previous studies). The authors also discovered that, in contrast to S. cerevisiae, both Pkh1 and Pkh3 play important roles in regulating Ypk1. However, weak points of the manuscript were that the endocytosis assay was not done in a sensitive manner and several assays were only presented in a qualitative manner that lacked quantitative conclusions. Altogether, this study is significant for identifying the roles of the conserved Ypk1 kinase in a major fungal pathogen.

1 Fig. 2A. It is not clear that myriocin has a greater effect on the ypk1D mutant just from looking at this spot assay. Myriocin affects the growth of both WT and ypk1D mutants, and since the ypk1D mutant grows slower the effects appear to be more dramatic. Therefore, quantitative assays should be done.

2. Fig. 2C. The experiment that is presented is not sufficient to support the conclusion that the ypk1D mutant does not have an endocytosis defect. The cells were stained with FM4-64 for a long time (30 min) and then cells were incubated for an additional 90 min. This very long incubation could easily mask even a strong endocytosis defect. In order to support the conclusion that the ypk1D mutant does not have an endocytosis defect, a shorter time of FM4-64 treatment should be used and then short times of incubation should be analyzed, such as 15 min and 30 min.

3. The growth rate of the wild type and ypk1D strains should be quantified to better understand how much slower is the growth of the ypk1D mutant.

4. Fig. 3B. Growth rates for ypk1D and ypk1D fpk1mutants should be quantified to support the conclusion that ypk1D fpk1 grows faster.

Fig. 3C should be quantified (even though the results looks convincing, this would strengthen the conclusion).

Figure 7. Ypk1 bands should be quantified.

Minor:

Figure 1 should be labeled with the genotypes in addition to strain numbers to make it less confusing.

**Have all data underlying the figures and results presented in the manuscript been provided?**

Reviewer #1: Yes

Reviewer #2: Yes

PLOS authors have the option to publish the peer review history of their article (what does this mean?). If published, this will include your full peer review and any attached files.

Reviewer #1: No

Reviewer #2: No

---

## [Decision Letter · Decision Letter 1]

28 Jul 2023

Dear Dr. Morschhauser,

We are pleased to inform you that your manuscript entitled "The Ypk1 protein kinase signaling pathway is rewired and not essential for viability in Candida albicans" has been editorially accepted for publication in PLOS Genetics. Congratulations! The reviewers were enthusiastic about the modifications/additions and agreed that it is a strong manuscript.

Yours sincerely,

Anna Selmecki, Ph.D.

Academic Editor

PLOS Genetics

Geraldine Butler

Section Editor

PLOS Genetics

Comments from the reviewers (if applicable):

Reviewer's Responses to Questions

**Comments to the Authors:**

Reviewer #1: The authors have addressed my criticisms well. I think that the summary figure is a very helpful addition for readers.

Reviewer #2: The authors have addressed my concerns by quantifying several experiments and carrying out shorter time course in the endocytosis analysis. The results now provide stronger support for the conclusions.

**Have all data underlying the figures and results presented in the manuscript been provided?**

Reviewer #1: Yes

Reviewer #2: Yes

PLOS authors have the option to publish the peer review history of their article (what does this mean?). If published, this will include your full peer review and any attached files.

Reviewer #1: No

Reviewer #2: No

**Data Deposition**

http://datadryad.org/submit?journalID=pgenetics&manu=PGENETICS-D-23-00363R1

**Press Queries**

---

## [Editor Report · Acceptance letter]

6 Aug 2023

PGENETICS-D-23-00363R1 

The Ypk1 protein kinase signaling pathway is rewired and not essential for viability in Candida albicans 

Dear Dr Morschhäuser, 

We are pleased to inform you that your manuscript entitled "The Ypk1 protein kinase signaling pathway is rewired and not essential for viability in Candida albicans" has been formally accepted for publication in PLOS Genetics! Your manuscript is now with our production department and you will be notified of the publication date in due course.

With kind regards,

Zsofia Freund

PLOS Genetics

On behalf of:
